# Short-Term Effect of Additional Daily Dietary Fibre Intake on Appetite, Satiety, Gastrointestinal Comfort, Acceptability, and Feasibility

**DOI:** 10.3390/nu14194214

**Published:** 2022-10-10

**Authors:** Erika Borkoles, Digby Krastins, Jolieke C. van der Pols, Paul Sims, Remco Polman

**Affiliations:** 1School of Medicine & Dentistry (Public Health), Griffith University, Gold Coast 4222, Australia; 2School of Exercise & Nutrition Sciences, Queensland University of Technology, Brisbane 4059, Australia; 3Centre for Agriculture and the Bioeconomy, Queensland University of Technology, Brisbane 4059, Australia; 4The Healthy Grain Pty, South Yarra, Victoria 3141, Australia

**Keywords:** barley, fibre, appetite, satiety, gastrointestinal, mixed method

## Abstract

*Background*: There is evidence that high-fibre diets have significant health benefits, although the effect of increasing fibre on individuals’ appetite, satiety, and gastrointestinal comfort is not well established, nor is its acceptability and feasibility. *Methods*: This mixed-methods feasibility randomised control trial included 38 participants allocated to one of three conditions: FibreMAX (two daily servings of 25 g of BARLEYmax^®^), FibreGRAD (two daily servings with the amount of fibre gradually increased), and Control (two daily servings totalling 25 g of placebo product). Participants completed a food diary at baseline. The Hunger and Fullness Questionnaire and questions regarding gastrointestinal response were completed at baseline and at the end of each week. Participants completed the acceptability of intervention measure and engaged in a semi-structured interview, following trial completion. *Results*: The qualitative data suggested that increased fibre influenced appetite and fullness perceptions. Baseline fibre consumption and the method of increased fibre increase did not influence our findings. The qualitative results also indicated that the fibre intake was perceived as beneficial to well-being; it influenced feelings of hunger and caused some minor acute gastrointestinal symptoms that dissipated after a short adaption period. *Conclusion*: This study suggests that increasing fibre intake through BARLEYmax^®^ is a safe intervention that is acceptable to participants.

## 1. Introduction

Obesity prevalence is still growing [1] and is a key risk factor for several chronic and non-communicable [2] diseases. It has contributed to much of the increase in the burden of disease globally [3] and was declared by the World Obesity Federation as a ‘chronic progressive disease’ [4]. Lifestyle interventions that could help combat the obesity epidemic and that are practical and easy to integrate into people’s daily routine would greatly reduce the burden of disease. The Western diet contributes to increases in the prevalence of obesity, as it is characterised by consuming energy-dense food that is of low nutritional value [5,6]. The general public health advice of limiting energy intake to reduce weight and improve body composition has not been a successful strategy, as obesity rates continue to increase [7]. Therefore, alternative food-consumption strategies need to be developed that can improve body weight and body composition, reduce the prevalence of lifestyle diseases, and increase well-being.

There is extensive evidence that a high-fibre diet is associated with a lower risk for certain cancers, heart disease, diabetes, stroke, and all-cause mortality [8,9]. Furthermore, a recent systematic review and meta-analysis of 62 human trials showed that supplementing viscous fibre influences body weight, body-mass index (BMI), and waist circumference independent of an energy-restrictive diet [10]. These benefits are seen after a median duration of 8 weeks (body weight and BMI) and 10 weeks (waist circumference) [11]. Additionally, the duration of the intervention appears to be important, as interventions that last longer than 8 weeks typically result in more significant weight loss [9]. A proposed mechanism of the benefit of fibre on weight control is the suppression of appetite and increased satiety [12,13]. Likely, as foods high in fibre are bulky and require more chewing, this results in feeling fuller and subsequently reduces energy intake, which may improve body composition.

While supplementing with fibre typically suppresses appetite and increases satiety resulting in weight loss, the fibre type is also necessary to consider, as some fibre types have minimal or a negative effect on appetite, satiety, energy intake, and weight loss [14]. Studies of fibre types such as β-Glucan (from barley) have shown that acute consumption suppresses appetite, increases satiety, and reduces energy intake [15]. In their editorial, Hjorth and Astrup [7] suggested that more studies are required to examine the effect of particular viscous fibre types on body composition to better understand the relationship between dietary fibres, satiation, and physiological processes.

Consumption of dietary fibre is also important for gut ecology. In particular, studies suggest that high-fibre diets are associated with microbiota richness. In contrast, Western diets low in fibre might result in the lasting disappearance of some microbial taxa [16]. Notably, the microbiome rapidly responds to extreme dietary changes (within 24 h) [17].

Similarly, increased dietary fibre can lead to changes in gastrointestinal function, primarily in the early stages of changing diet [18]. This includes increased flatulence [19], stool change, abdominal discomfort (e.g., bloating, cramping, and feeling too full), or changed bowel habits (e.g., urgency of bowel movements and problems with defecation) [20]. However, few studies have examined how initial gastrointestinal discomfort might change over time. Gastrointestinal comfort is important in ensuring the feasibility, acceptance, and appropriateness of studies examining the efficacy of increased dietary fibre intake on long-term weight loss. For example, knowing how to minimize gastrointestinal discomfort with a high-fibre diet can prevent dropout in studies and improve long-term adherence.

Therefore, we investigated how three weeks of increased fibre intake would affect participants’ appetite, satiety (feelings of hunger and fullness), and gastrointestinal comfort. We also investigated whether potential moderators influence the response to increased fibre intake (e.g., baseline fibre consumption and gradual increase of fibre intake), and whether increased fibre intake is acceptable, appropriate, and feasible to participants and the best method of consumption (e.g., which meal in combination with which foods). We hypothesised that increasing fibre intake with a barley-derived supplement would lead to decreased appetite and increased satiety. Additionally, we hypothesised that there would be some early negative consequences on participants’ gastrointestinal comfort, which would dissipate over the study period.

## 2. Materials and Methods

### 2.1. Participants

This study recruited 38 participants between 18 and 50 years of age with a BMI of 18.5 to 34.9 kg·m^−2^ (normal to obese BMI) from the Brisbane area. See Table 1 for participants’ characteristics in the three study conditions (see inclusion/exclusion criteria in procedure section). Participants provided written consent, and the study was approved by a university human ethics committee and registered in the Australian and New Zealand trial register (ACTRN12621001440819).

### 2.2. Study Materials

Participants completed several questionnaires at baseline and at the end of week 1, 2, and 3 of the study. The following instruments were used.

Food questionnaire: Participants completed the National Health and Nutrition Examination Survey (NHANES) dietary screener questionnaire at baseline to determine their fibre intake. This questionnaire is a 26-item recall questionnaire that asks about the frequency of consumption of selected food and drinks in the last month.

Hunger and Fullness Questionnaire: The 5-factor Hunger and Fullness Questionnaire (HFQ) developed and validated by Karalus et al. [21] was used to assess satiety. The five factors are mental hunger, physical hunger, mental fullness, physical fullness, and food liking. The questionnaire is scored on a 10-point scale with a not appropriate response option. The HFQ was mainly developed to examine products that are likely to provide greater satiation and satiety [22]. The HFQ was sent to participants weekly and completed at baseline and each subsequent week (4 times in total).

Gastrointestinal response and stool consistency: The frequency of bowel movements was recorded, and the gastrointestinal response was assessed weekly based on questions developed by Chen et al. [23]. The ease of bowel movement was graded as 1 (easy), 2 (slightly difficult), 3 (difficult), and 4 (extremely difficult). Feelings of complete relief were graded as 1 (extremely agree), 2 (agree), 3 (disagree), and 4 (extremely disagree). The side effects, such as abdominal cramping, stomach rumbling (borborygmi), bloating, and flatulence (breaking or passing wind), were graded a 0 (no symptom), 1 (very slight symptom), 2 (slight symptom), 3 (severe symptom), and 4 (very severe symptom). The stool consistency was graded as 1 (very hard), 2 (hard), 3 (soft), 4 (very soft), and 5 (watery) whenever bowel pass occurred.

Acceptability, appropriateness, and feasibility of increased dietary fibre: The acceptability of intervention measure (AIM), intervention appropriateness measure (IAM), and feasibility of intervention measure (FIM) [24] were used to examine whether the increase of dietary fibre was acceptable, appropriate, and feasible to participants. There are 4-items for each factor which were modified to relate to the present study context. Items are scored on a Likert-scale form: 1 = completely disagree, 2 = disagree, 3 = neither agree nor disagree, 4 = agree, and 5 = completely agree. The AIM, IAM, and FIM have been shown to have strong psychometric properties, including good validity, reliability, and structural invariance [24].

### 2.3. Study Procedure

Participants in this feasibility RCT were recruited through advertisements on Facebook, in local advertising, and in emails. Interested participants who contacted the research team were then screened for eligibility to the study. The inclusion criteria were being aged between 18 and 50 years of age, having a BMI between 18.5 and 34.9 kg·m^2^, and being located in the greater Brisbane area. People were excluded from participating in the study if they had a history or presence of a comorbid disease for which diet modifications may be contraindicated (e.g., diabetes, metabolic syndrome, polycystic ovary syndrome, and hypoproteinemia); a history of eating disorders; a history of bariatric surgery; or were pre-menopausal, menopausal, pregnant, or breastfeeding. People were also excluded if they had taken dietary supplements for thyroid, hyperlipidemia, hypoglycaemia, or weight loss, or had participated in a diet or clinical trial in the last six months. People were also not eligible if they had a mental or cognitive impairment that limited their ability to understand test instructions.

This was a mixed-method, feasibility randomised controlled trial. Participants were randomly assigned to one of three conditions: (1) receive two 25 g daily servings of a barley-derived fibre product (12.7 g fibre per 100 gr product; BARLEYmax^®^ South Yarra, Victoria, Australia) (FibreMAX condition); (2) receive two daily servings of the barley product, where the amount was gradually increased over the 3-week period (week 1 = 8 g per serving; week 2 = 16 g per serving; week 3 = 25 g per serving) (12.7 g fibre per 100 gr product; FibreGRAD condition); (3) receive two 25 g daily servings of a placebo product (wheat flakes, 6.4 g fibre per 100 gr product) (Control condition).

The additional fibre in the form of BARLEYmax^®^ and the placebo was provided to the participants in the form of flakes packed in a foil sachet. The BARLEYmax^®^ product is a whole-grain barley that contains 25.5 g of fibre per 100 g (including 6.4 g of β-glucan and 2.6 g of resistant starch). The control wheat flakes used were wheat bran flakes (Uncle Tobys^®^ Weeties, Wahgunyah, Victoria, Australia). The placebo was a good match in terms of taste, colour, and texture. Additionally, the flakes used were 99% whole wheat. The placebo wheat flakes consisted of 12.7 g of fibre per 100 g of product. Typically, wheat is considered to have the lowest content of β-Glucan among various grain tissues (0.4–1.4%) [25]. The participants were not provided with information on how or when to consume the dose, other than that this should be on two separate occasions (e.g., breakfast and lunch; breakfast and dinner; or lunch and dinner). There was no information on the packaging, other than the letter A, B, or C (referring to the condition), limiting participants from guessing their condition allocation. Additionally, while not listed on the package, the packages had different weights (8 g and 25 g). Considering that satiety and gastrointestinal functioning are influenced by supplementation almost immediately, a 3-week intervention period was deemed appropriate.

Participants attended an initial assessment visit, completed questionnaires each week for three weeks, and attended a final face-to-face evaluation interview. During the first visit, screening information was obtained to determine participants’ baseline characteristics and ensure participants’ eligibility. Additionally, participants completed the dietary screener to assess their current fibre intake. When participants were deemed eligible and consented to participate in the study, they were randomized into one of the three conditions by using envelopes. The researcher was not blind to the condition.

At baseline and at the end of each week (four times total), a short set of questionnaires was used to determine the effect of the intervention on participants’ reported satiety and appetite (HFQ), gastrointestinal response, and compliance to the study. At the end of week three, the participants were also asked to complete the AIM, IAM, and FIM. During the final visit, a study evaluation interview was conducted to gain an understanding of the participants’ experience of taking the fibre flakes provided, the study protocol in general, and their compliance. The interviews length averaged from 30 to 45 min to assess the acceptability, appropriateness, and feasibility of consuming additional fibre in each trial condition.

### 2.4. Statistical and Qualitative Data Analysis

No power calculation was conducted for this study [26]. The sample size for this study was based on recommendations of between 24 and 50 participants [27,28] or 12 per group [28]. To inform the estimation of the SD for use in the sample size calculation for a subsequent RCT, we decided to have up to 15 participants per group, expecting a 20% dropout rate.

In the first instance, repeated-measures analysis of covariance (ANCOVA) was used to analyse the primary and secondary outcome data. The independent factor was the study condition (FibreMAX, FibreGRAD, and Control), and we used the fibre intake at baseline as a covariate. In the instance of a significant interaction effect, we used a t-test to explore group differences. Sidak post hoc comparisons were used to explore condition or time main effects. In the instance of the covariate not influencing results, we conducted repeated-measures ANOVA. Finally, MANOVA was used to explore differences between the conditions for acceptability, appropriateness, and feasibility factors, and ANOVA was used for baseline fibre intake. Effect sizes for ANOVA are reported as partial eta squared (ηp2) with a small effect at 0.010–0.059, a medium effect at 0.060–0.139, and a large effect > 0.14. Effect sizes for t-tests are reported as Cohen’s d with a small effect at 0.20–0.49, a medium effect at 0.50–0.79 and a large effect > 0.80. Prior to data analysis, data were scanned for normality, outliers, and homoscedasticity. The qualitative data (interviews) were transcribed verbatim and analysed by using a deductive thematic analysis [29]. This analysis focused on the acceptability and feasibility of the use of the product. The analysis was performed by the first author initially and then checked by the last author for accuracy. Example questions included ‘could you tell me about your experience of increasing fibre in your diet through the flakes provided?’ and ‘how closely have you followed the use of the flakes as recommended?’ The participants gave a description of symptoms for week 1, 2, and 3 and their adherence to the trial. Note, the interviewer was blinded to the participants’ condition.

## 3. Results

Of the 38 participants initially recruited, 32 completed the study (FibreMAX, *n* = 10; FibreGRAD, *n* = 9; Control, *n* = 13) (see Figure 1). Four participants were lost to follow-up (FibreMAX, *n* = 2; FibreGRAD, *n* = 1; Control, *n* = 1), two participants withdrew due to adverse symptoms (FibreMAX, *n* = 1; Control, *n* = 1), and one participant withdrew due to being too busy (FibreGRAD, *n* = 1). The data did not violate any assumptions of normality or homogeneity of variance. The baseline fibre intake did not influence any of the findings. As such, the results of the repeated-measures ANOVA are reported.

### 3.1. Appetite and Satiety

Table 2 provides the results of the repeated-measures ANOVA, and Figure A1a–e in Appendix A show the five factors related to appetite and satiety (mental hunger, physical hunger, mental fullness, physical fullness, and food liking). There were no main or interaction effects, and the effects size was low to medium.

### 3.2. Gastrointestinal Response—Quantitative Survey Data

Table 2 also provides the results of the repeated measures ANOVA for the gastrointestinal responses, whereas Figure A2a–f in Appendix A provide the mean scores of participants in the different conditions across the four measurement points. There was a time main effect for an increase in breaking or passing of wind and hardening of stool consistency, both with a large effect size. Post hoc comparisons, however, did not show any significant differences between the three conditions.

### 3.3. Gastrointestinal Response—Qualitative Interview Evaluation

Table 3 details the frequency of side effects per week per condition. During the evaluation interviews, each participant was asked to indicate weekly whether she/he experienced any side effects and what those specific side effects were. The frequency data indicate that it took a week for participants to adjust to the increased fibre intake, and they mainly reported negligible side effects in weeks 2 and 3 of the trial. In the FibreMAX group, after week 1, participants reported eating less overall and sometimes used the extra fibre as a meal replacement. In the FibreGRAD condition, one theme that emerged was that participants reported feeling too full throughout the trial period.

### 3.4. Study Evaluation: Acceptability, Appropriateness, and Feasibility

There were no significant differences between the three conditions regarding the acceptability, appropriateness, and feasibility of increasing dietary fibre (MANOVA Wilks lambda = 0.89, P = 0.84; ηp2 = 0.05). Table 4 provides information on the responses by the participants in the three conditions regarding the acceptability, appropriateness, and feasibility of the study.

### 3.5. Results of the Evaluation Interviews

#### 3.5.1. Product Evaluation

When asked to describe the product, participants in the FibreMAX and FibreGRAD conditions used some of the following words: ‘sour’, ‘crunchy’, ‘chewy’, ‘unpalatable’, ‘tasting like a cardboard’, ‘didn’t smell’, ‘dry’, ‘grainy taste’, ‘no taste’, ‘bland’, ‘yuck’, ‘rough’, ‘feels like medicine, rather than food’, and ‘odd to eat’. Despite some of the product descriptions, overall, participants were satisfied with the additional fibre, especially when mixed and consumed with other food items. However, some were frustrated with using the product because they saw the product as a breakfast cereal and could not think of how else they would enjoy consuming it (i.e., lack of palatability). They also found it challenging to adhere to the study protocol during weekends, as their weekend routine was different from their weekday routine. In general, participants reported that they would prefer to consume the product as a breakfast item or a snack.

Participants in the control conditions described the product as ‘very sweet’, ‘yummy’, ‘nice’, ‘tastes like Weet-bix’, and ‘crunchy’. The description of the wheat-based product was less rich than that of the barley products.

Participants were asked to rate the acceptability and ease of consuming the product. The mean scores for ‘acceptability’ and ‘ease’ of consuming the product were 8 out of 10 in both the FibreMAX and FibreGRAD groups. In the control group, the mean ‘acceptability’ score was 9, and the ‘ease’ to consume was 10.

Participants also commented on the appearance of the packaging of the product, wanting more information on how the flakes could be used in different ways to avoid boredom (e.g., recipe book or videos), as well as different ways of providing additional fibre (other than just flakes, such as a breakfast bar). Moreover, the social acceptability of consuming the product was low overall. Participants reported being uncomfortable consuming the product in social situations, citing stigma-related social stress, such as being judged as ill or being weird. Participants also indicated that they considered internal cues above external cues of food environment when experimenting with a new way of eating, especially sensory cues, such as smell, texture, look, and the product’s attractiveness.

#### 3.5.2. Food Preferences

Participants in the FibreMAX and FibreGRAD conditions preferred the product to be palatable and healthy.

#### 3.5.3. Hunger, Satiety, and Cravings

Participants in the FibreMAX and FibreGRAD reported feeling ‘overly’ full following consumption of the product (91% FibreMAX; 100% FibreGRAD). ‘Overly full’ is a description associated with reducing overeating. Surprisingly, also 10 control participants reported being full during week 1 but not during the subsequent weeks. In addition, participants in the FibreMAX and FibreGRAD conditions reported less snacking, more delaying of next food intake, and reduced the portion size of their subsequent meals, as they did not feel hungry.

#### 3.5.4. Anecdotal Health Benefits

Although it does not indicate causality, two participants reported that they received health benefits from the increased fibre intake. One participant (FibreMAX) with a respiratory disease reported a decrease in ventilator use and a reduction in anti-inflammatory medication intake despite a concurrent period of high smoke haze in the bushfire season. Another participant from the FibreMAX group who had a pre-existing bowel condition reported a significant decrease in bowel problems and indicated improved stool consistency. Although both attributed the improvements to increased fibre intake, it is unclear whether this is the case. Moreover, in the FibreMAX group, four other female participants reported healthier bowel movements around their monthly cycles. Eighty per cent of the participants reported some benefit to their bowel movements and gut health due to being in the study.

#### 3.5.5. Side Effects

As per Table 3, the FibreMAX and FibreGRAD participants reported some adverse ‘side effects’ of the additional fibre consumption. The two main complaints were flatulence and abdominal discomfort (bloating and cramping) and were rated as moderate to severe. In the FibreMAX condition, 25% reported flatulence and 73% abdominal discomfort, but these two side effects were connected. In the FibreGRAD condition, this was 50% and 63%, respectively. In addition, participants in the FibreGRAD condition reported more side effects in week 3 when the maximum fibre dose of 50 g was consumed. Surprisingly, 29% of participants in the control group reported some mild abdominal discomfort. On the other hand, for the FibreMAX condition, side effects were reduced significantly in 90% of participants during week 3. This might have been due to adaptations to the product.

The mean values for likeliness to continue consuming the products for participants in the FibreMAX condition were 4 out of 10, but this was 8 out of 10 in the FibreGRAD condition in weeks 1 and 2, and 5 out of 10 for week 3 of the trial.

Participants in all conditions reported changes in stool function, and this was mainly evaluated as a positive change. Participants reported a significant increase in stool volume and change in its quality, in line with previous findings [30]. In general, participants reported that they had easier bowel movements and felt healthier after having more regular defecation. Participants also reported more acceptable stool consistency.

By the end of the trial, participants used the product as a ‘meal replacement’, because they felt too full. This also reportedly resulted in reduced snacking behaviours and eating smaller portion sizes. In addition, participants in the barley conditions reported having delayed their subsequent meal by 2 h, on average, before they were ready to eat again (e.g., time between breakfast with product and lunch).

#### 3.5.6. Quality of Life and Psychological Well-Being

Participants in all groups reported perceived health benefits of participating in the study and reported feeling well, less tired, and able to do more in their daily lives, except for week 3 in the FibreGRAD condition, where participants felt too full and reported being sluggish and tired. This was also the case for participants in the FibreMAX condition in week 1, when they were adapting to the increased fibre intake. In the control condition, the participants also reported increased well-being and being more energetic by the end of the study.

#### 3.5.7. Compliance with Study Protocol

All 38 participants were interviewed and checked for compliance. All reported the easy use of sachets, noted the convenience of mixing it with other food products, and praised the portability of the product. Participants used the product as per prescription. Self-reported compliance was 95%. Some sachets were not consumed mainly due to forgetting to take them to work or were outside of the home.

## 4. Discussion

In this study, we investigated whether three weeks of increased fibre intake would influence appetite and satiety (feelings of hunger and fullness); participants’ gastrointestinal comfort; whether potential moderators influence the response to increased fibre intake (e.g., baseline fibre consumption and gradual increase of fibre intake); and whether increased fibre intake is acceptable, appropriate, and feasible to participants and the best method of consumption (e.g., which meal in combination with which foods). Our results suggest that three weeks of increased fibre intake (a) does not affect appetite or satiety, as measured by questionnaires, whereas qualitative data suggested that it influenced participants’ appetite; and (b) it increases the frequency of breaking or passing wind and increases stool consistency. Additionally, the baseline fibre consumption and the method of increased fibre increase (gradual or full) did not influence the quantitative findings. The qualitative results indicated that the consumption of additional fibre was perceived as beneficial to overall well-being, influenced feelings of hunger, and caused some minor gastrointestinal symptoms, which dissipated after a short adaptation period.

Feelings of hunger and fullness play an important role in obesity management [31]. In Australia, overeating has been identified as one of the main reasons for this rise [32]. There are many socio-ecological reasons for why overeating occurs, including physiological factors (e.g., hunger and satiety) [33] and environmental factors, such as availability and low price of high energy density foods, large portion sizes, increased flavouring for palatability, and liquid versus solid state food consumption [34].

In this study, with three weeks of fibre supplementation, there was no change in appetite or satiety, as measured by the HFQ. Importantly, and contrary to the quantitative findings, the interviews suggested that participants in both fibre supplementation conditions felt fuller and had a reduced appetite. Therefore, the qualitative findings of this study were a more sensitive measure of hunger and satiety in this short feasibility study, showing that increased intake of a high-fibre product derived from barley can reduce hunger and improve satiety feelings. This would support previous findings that high fibre intake can play a role in preventing overeating, reducing snacking behaviours, and improving health outcomes [35]. The findings of the present study have provided some important recommendations for future studies by using increased dietary fibre to achieve health benefits (e.g., weight loss).

The non-change in the HFQ was unexpected, considering that fibre from barley sources has been associated numerous times with decreased appetite and reduced self-reported food intake [36]. The current study used a different measurement technique that may not be sensitive enough to detect changes in appetite and satiety in this short-duration study. Additionally, satiety fluctuates throughout the day, particularly around mealtime, and is largely dependent on the individuals physical and psychological characteristics. As such, examining satiety in a field study like this is more difficult. The accuracy of findings could be increased by measuring participants’ satiety levels prior to and following a specific meal. However, such experimental control reduces ecological validity, particularly in studies that last a relatively long time. When looking at the data from the interviews, it appears that the fibre supplementation did influence the participants’ feeling of fullness, thus further supporting that the measurement tools may not have been sensitive enough, or that the standardisation of completing the tool should have been more rigorous. The feelings of increased satiety might be due to stomach distention, fermentation, and changes in gut hormones [36]. The findings from the interviews are particularly important, as there are few solutions to reduce overeating [37]. These findings would explain how individuals could gain control over their eating habits, thus helping them manage their weight and improve their health. This would be particularly beneficial for overweight and obese individuals, i.e., higher satiety, lower craving, and reduced overeating and snacking.

After three weeks of fibre supplementation, the data suggested an increase in the frequency of breaking or passing wind and hardening of stool consistency in both fibre-supplementation conditions. There was no indication that other side effects, such as bloating or cramping, increased in the fibre conditions. This indicates that, for the most part, participants did not experience significant negative gastrointestinal responses due to the increased fibre intake. The interview data highlighted that the participants generally felt well. However, when consuming 50 g of additional fibre daily (FibreMAX W1 and FibreGRAD W3), there was some abdominal discomfort and flatulence that took 3 to 7 days to adapt. Together, these results were in line with previous research that indicated that increased fibre consumption is associated with increased flatulence, bloating, and uncomfortable abdominal distension [38], and a trend for changes in stool consistency has been previously found [39]. It appears that it is important to consider participants’ short-term adaptation to changes in fibre intake for the feasibility of studies and to increase the fibre intake gradually. Long-term fibre supplementation appears to improve gastrointestinal health, as increased fibre intake is associated with mitigating and reducing gastrointestinal symptoms, which improves quality of life [40]. It appears that discomfort is temporary, and longer-term supplementation has the potential to result in positive health changes [16]. However, more research on longer-term duration interventions is needed to confirm this.

Overall, the mixed-method results indicated that the participants found the trial acceptable, appropriate, and feasible. In terms of acceptability and appropriateness, participants agreed that increased fibre in their diet was manageable. However, if they had to take it for a longer duration, they indicated that the flakes used in the present study would require some adaptation to their needs, such as consuming the product in different forms (e.g., cereal bar and trail mix). They also wanted to be provided with more elaborated recipes and to be shown how the flakes could be consumed in different ways. Finally, in terms of feasibility, our results clearly indicate that it is possible to increase fibre in an individual’s diet by supplementing products like the one used in the current study.

The findings of this study were limited by a relatively small sample size, and this may explain the lack of significant findings in the quantitative comparisons. The present study was designed as a feasibility study and was not designed to achieve sufficient statistical power to detect group differences. The intervention for this trial also occurred primarily throughout December, a time of the year in which some participants may have changes in diet that are unrelated to the supplementation. The surveys were sent to participants weekly, and the time of day they were completed was not controlled; this may have decreased the sensitivity to changes throughout the intervention. The assessment of fibre intake used an instrument that provided a rough estimate and only assessed dietary intake at one point in time.

## 5. Conclusions

This study suggests that increasing fibre intake through BARLEYmax^®^ is a safe intervention that presents no major risks. The qualitative results also suggest that a gradual increase might be more beneficial in future trials to avoid early dropout from studies due to gastrointestinal discomfort. However, further research that analyses the longer-term effects of increased dietary intake of high-quality fibre is needed to determine its effect on appetite, satiety, and body composition.

## Figures and Tables

**Figure 1 nutrients-14-04214-f001:**
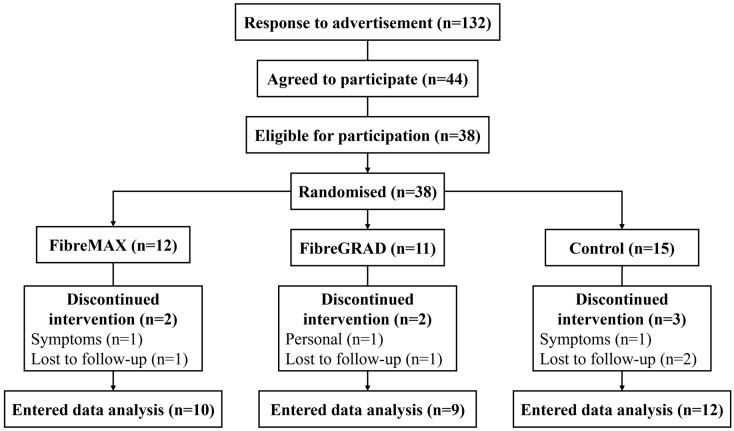
Consort diagram of study flow.

**Table 1 nutrients-14-04214-t001:** Participant characteristics by study condition (values mean ± SD or frequency distribution in percentage) and the results of the one-way ANOVA (F value and *p*).

	FibreMAX *n* = 12	FibreGRAD *n* = 11	Control *n* = 15	F(2,35) & *p*
Age, years	32 ± 8	30 ± 10	32 ± 9	0.19; *p* = 0.83
Male gender, *n* (%)	3 (25)	1 (9)	5 (33)	
Weight, kg	78 ± 22.2	63.1 ± 7.6	69.9 ± 12.3	2.71; *p* = 0.08
Height, cm	170.1 ± 12.4	165.2 ± 5.9	168.5 ± 11.9	0.63; *p* = 0.54
Body mass index, kg·m^−2^	26.6 ± 4.2	23.1 ± 2.3	24.6 ± 4.1	2.58; *p* = 0.09
Baseline fibre intake (gr)	17.0 ± 4.0	17.6 ± 4.0	16.7 ± 3.4	0.21; *p* = 0.81

**Table 2 nutrients-14-04214-t002:** Results of the repeated measures analysis of variance for the five satiety factors, namely mental hunger (MH), physical hunger (PH), mental fullness (MF), physical fullness (PF), and food liking (FL), and the gastrointestinal response (EBM = ease of bowel movement; BL = bloating; AC = abdominal cramping; Wind = breaking or passing wind; STR = stomach rumbling (borborygmi); SC = stool consistency).

Satiety	Condition Main Effect (F(2,28))	Time Main Effect (F(3,81))	Condition × Time Interaction (F(6,81))
MH	0.41; *p* = 0.67; ηp^2^ = 0.03	2.11; *p* = 0.11; ηp^2^ = 0.07	0.42; *p* = 0.87; ηp^2^ = 0.03
PH	0.78; *p* = 0.47; ηp^2^ = 0.06	0.98; *p* = 0.41; ηp^2^ = 0.04	0.59; *p* = 0.73; ηp^2^ = 0.04
MF	0.88; *p* = 0.51; ηp^2^ = 0.06	0.57; *p* = 0.64; ηp^2^ = 0.02	0.08; *p* = 0.92; ηp^2^ = 0.01
PF	0.61; *p* = 0.55; ηp^2^ = 0.04	0.80; *p* = 0.50; ηp^2^ = 0.03	0.60; *p* = 0.73; ηp^2^ = 0.04
FL	0.97; *p* = 0.39; ηp^2^ = 0.07	0.67; *p* = 0.57; ηp^2^ = 0.02	0.11; *p* = 0.99; ηp^2^ < 0.01
Gastrointestinal			
EBM	0.18; *p* = 0.84; ηp^2^ = 0.01	0.18; *p* = 0.91; ηp^2^ = 0.01	0.17; *p* = 0.98; ηp^2^ = 0.01
BL	0.40; *p* = 0.67; ηp^2^ = 0.03	1.29; *p* = 0.29; ηp^2^ = 0.05	0.80; *p* = 0.58; ηp^2^ = 0.06
AC	1.71; *p* = 0.20; ηp^2^ = 0.11	1.31; *p* = 0.28; ηp^2^ = 0.05	0.29; *p* = 0.94; ηp^2^ = 0.02
Wind	1.18; *p* = 0.32; ηp^2^ = 0.08	3.63; *p* = 0.02; ηp^2^ = 0.12	1.46; *p* = 0.22; ηp^2^ = 0.10
STR	0.77; *p* = 0.47; ηp^2^ = 0.05	1.91; *p* = 0.14; ηp^2^ = 0.07	0.65; *p* = 0.69; ηp^2^ = 0.05
SC	0.09; *p* = 0.91; ηp^2^ < 0.01	2.80; *p* = 0.04; ηp^2^ = 0.10	0.45; *p* = 0.84; ηp^2^ = 0.03

**Table 3 nutrients-14-04214-t003:** Frequency of reported side effects for each condition per week.

	FibreMAX (*n* = 11)			FibreGRAD (*n* = 8)			Control (*n* = 14)		
Side Effects	W1	W2	W3	W1	W2	W3	W1	W2	W3
Gastrointestinal discomfort	6	0	0	1	1	1	1	0	0
Flatulence	4	0	0	4	4	4	8	0	0
Stool change	4	0	0	5	0	0	7	0	0
Abdominal discomfort	8	0	0	2	3	2	1	0	0
Bloating	6	0	0	5	5	5	5	0	0
Cramping	4	0	0	3	3	3	3	0	0
Feeling too full	10	0	0	8	8	8	10	0	0
Meal replacement	0	0	0	5	0	0	0	0	0
Changed bowel habits	11	0	0	2	0	0	0	0	0
Urgency of bowel movements	4	0	0	2	3	3	2	0	0
Problems with defecation	2	0	0	0	2	3	2	0	0
Dehydration	4	0	0	0	1	1	1	0	0
Nausea	0	0	0	1	1	1	0	0	0
Diarrhoea	0	0	0	2	0	0	0	0	0
Constipation	0	0	0	0	0	2	0	0	0
Stomach rumbling	0	0	0	3	0	0	0	0	0
Easier bowel movement	1	0	0	2	0	0	0	0	0

**Table 4 nutrients-14-04214-t004:** Results of the survey assessing participants acceptability, appropriateness, and feasibility with the MAXtrial. Values are mean ± SD. Scores: 1 = completely disagree, 2 = disagree, 3 = neither agree nor disagree, 4 = agree, and 5 = completely agree.

		FibreMAX *n* = 10	FibreGRAD *n* = 9	Control *n* = 12
Acceptability	Increasing fibre in my diet through these flakes meets my approval	3.60 ± 1.17	3.89 ± 0.93	3.67 ± 1.07
	Increasing fibre in my diet through these flakes is appealing to me	3.30 ± 0.82	2.78 ± 1.20	3.58 ± 1.16
	I like more fibre in my diet through these flakes	3.20 ± 1.14	3.33 ± 1.22	3.42 ± 1.16
	I welcome more fibre in my diet through these flakes	3.10 ± 1.29	3.56 ± 1.24	3.58 ± 1.08
Appropriateness	Increasing fibre in my diet through these flakes seems fitting	3.30 ± 1.06	3.56 ± 1.01	3.67 ± 0.98
	Increasing fibre in my diet through these flakes seems suitable	3.30 ± s1.06	3.44 ± 1.01	3.67 ± 1.07
	Increasing fibre in my diet through these flakes seems applicable	3.20 ± 1.14	3.33 ± 1.12	3.67 ± 1.23
	Increasing fibre in my diet through these flakes seems like a good match	3.20 ± 1.23	3.33 ± 1.00	3.50 ± 1.45
Feasibility	Increasing fibre in my diet through these flakes seems implementable	3.70 ± 0.48	3.25 ± 1.16	3.58 ± 1.00
	Increasing fibre in my diet through these flakes is possible	3.60 ± 0.70	3.75 ± 0.71	3.82 ± 0.60
	Increasing fibre in my diet through these flakes seems doable	3.60 ± 0.70	3.75 ± 0.71	3.36 ± 1.03
	Increasing fibre in my diet through these flakes seems easy to use	3.60 ± 0.52	3.13 ± 1.13	3.25 ± 1.06

## Data Availability

Data are available on request from the corresponding author. The data are not publicly available due to commercial interest.

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
