# Peer review of "Short-Term Effect of Additional Daily Dietary Fibre Intake on Appetite, Satiety, Gastrointestinal Comfort, Acceptability, and Feasibility"

_nutrients, 2022, doi:10.3390/nu14194214_

Round 1

Reviewer 1 Report

This manuscripts aims to elucidate how increased fibre-intake with the product FibreMax during 3 weeks affects appetite, satiety and gastrointestinal discomfort in a heterogenous Australian population. Dietary fibres might be a possible therapeutic intervention for people with obesity. Therefore, information concerning the acceptance of such an intervention is of potential interest. 

However, there are several major concerns: the study population is very small and very heterogenous in terms of age, sex and BMI which makes it very hard to draw any conclusions for the patient-population of interest, which is obese patients. No statistical analysis of the baseline characteristics is provided and it is not clear, whether subjects and investigators were blinded concerning the study arm. Major information concerning the study product and the placebo is missing (ingredients, kcal etc.). There is also no information on how the diet and anthropometric data changed during the intervention period.

With these flaws, the study is of limited relevance to the reader. 

Detailed response:

Introduction: 

- the current literature recommends at least 8 weeks of intervention. Why was the study performed for 3 weeks? Please elaborate.

- When talking about the potential mode of action of dietary fibres and weight loss, the microbiome and metabolism needs to be discussed

- The authors state that they hypothesize "without negative consequences on participants’ gastrointestinal comfort" even though they also state "Increased dietary fibre can lead to some changes in gastrointestinal function". Please elaborate.

Materials and Methods

- Were there significant baseline characteristics differences? Please add statistics.

- Wide range of BMI of the included patients could dilute results (heterogeneous study group, especially for such a small sample size), mean BMI were non-obese for the FibreGRAD and the control group and obese for the FibreMAX group. This is a major difference at baseline. 

- Placebo product wheat flakes: Very ill defined. Please state the dietary fibre content and dietary fibre types. Why is this an appropriate placebo? Please elaborate.

- "There was no other nformation on the packaging other than the letter A, B, or C." -> was this a double-blinded study? Please state and explain. 

- Please provide information about the randomisation process.

Results:

- Table 4: Avoid repetitive text and improve readability

- Section 3.5.3: "none of the 14 participants in the Control group felt ‘too full’ " contradictory to table 3 where 10/14 patients reported "feeling too full" as a side effect in week 1

- Section 3.5.4: Can any conclusions be drawn if 1 participant of the intervention group shows an improvement of a condition, but no one in the control group had the same condition?

-Section: 3.5.7: "All participants indicated high levels of compliance with the study protocol." Please elaborate, give some numbers.

- In general: Distinguish better between the control and the two intervention groups in the results section (e.g. "Participants reported a significant increase in stool volume and change in its quality, in line with previous findings" -> all participants?)

- Appendix: no information about the data shown (mean, median etc.) and no statistics. This is inappropriate. 

Discussion:

- Distinguish between effects observed in the two intervention groups and the control groups. Is there a significant difference? Are there hints that gradual increase of dietary fibre consumption is better compared to 50g starting from day 1?

- Scientific soundness of "two participants in the FibreMAX condition reported significant positive health benefits, including less bowel discomfort than they experienced before the trial."?

- Line 392: "It appears that discomfort is temporary and is likely to lead to positive changes with long-term supplementation." Discomfort is likely to lead to positive changes? Please revise. 

Conclusion: 

- Line 416: Add major or similar: no "major" risk. 

Author Response

Please see the attached file for our response to reviewer 1.

Reviewer 2 Report

In this paper, the authors conduct a randomized controlled trial to assess the effect of a fiber supplement on key outcomes.

Although the authors clearly identify this as a feasibility study, the small sample size and nebulous qualitative methods limit the contributions provided by this study. The authors clearly cite the benefits of fiber and beta glucan, so the novel contribution of this study needs to be more clearly explained. Furthermore, a more detailed description of the qualitative methods and results is necessary to support the conclusions of the study.

-L65. The authors state limitation is that few studies look at how GI discomfort changes over time; their study lasts 3 weeks. Please provide literature showing how long adaption takes to justify the 3-week intervention.

-A moderation effect seems unlikely with such a small sample size. Although the authors report no results, I believe the paper would be clearer by not even mentioning it.

-L96. Please insert a reference for validation of the NHANES tool.

-107. Please describe the use of the timing of the response measure in more detail (for example, were participants instructed to complete at the same time each day at the end of the week, in relation to a meal? Etc.)

-What was the fiber content of the wheat flakes control?

-Why did the sachets have different weights? This was unclear. Please explain.

-Please provide more detail about the qualitative methods, such as the questions that were asked. Also, please provide more details about the thematic analysis; for example, did multiple researchers code independently, or was it done by a single researcher?

-L265. The statement about BMI and palatability does not have data to support it. For example, what was the % of obese participants that made these comments? How many people does this represent?

-Section 3.5.4. This study was not designed to assess the impact on these specific health conditions; a single report of improvement in symptoms is insufficient to draw conclusions. To me, this section weakens the paper due to the lack of support; I recommend removing it.

-L337. Because the placebo group also reported increased perceived health benefits and feeling well, it seems misleading to say consumption of fiber was perceived as beneficial to well-being.

-L348. I disagree with the statement that the qualitative findings were more sensitive. There are other factors that could have influenced the outcomes. More details are needed about the qualitative methods and results in order to support the conclusions made in this paper overall.  

-L416. Minimal risk instead of no risk.

Author Response

Please see attached file for our response to reviewer 2

Round 2

Reviewer 1 Report

The testing of alternative dietary fibre sources is highly relevant as daily requirements are often unsufficiently met. However, there are several flaws in the study design which cannot be improved by simple editing of the manuscript text:

·       Inappropriate choice of placebo product: The aim of the study was “additional daily dietary fibre intake”, but the placebo product also contained 12.7 g dietary fibre per 100 g. I would not expect any major differences in neither the benefits nor the side-effects when comparing a daily dietary fibre doses of 12.7 g (FibreMax) vs. 6.4 g (Placebo).

·       In my eyes, this is more a comparison of two different dietary fibre products.

Some flaws can be improved by simple text edits:

·       Line 11-12: “We examined the effect of 3-weeks of increased fibre intake on appetite, satiety and gastrointestinal comfort.” This is rather the aim of the study than the background. Please state acceptability and feasibility as a primary outcome  in the abstract as well.

·       Line 64: “ Notably, the microbiome rapidly responds to dietary changes (within 24 hours) [17].” I would suggest to add “extreme dietary changes” 

·       Line 143: Please state daily dietary fibre dose, which I would see as the relevant information there: 12.7 g in FibreMax vs. 6.4 g in Placebo.

·       Line 158: “The placebo wheat flakes consisted of 158 12.7g of fibre per 100g of dietary fibre.” Please revise.

·       Line 301: This section still implies health benefits which are far-fetched without scientific soundess. 

Author Response

We have made the following changes based on the feedback provided by reviewer 1:

  • We have made changes to the opening sentences of the abstract as suggested.
  • Included extreme in line 64
  • Included fibre dose in this section.
  • Changed sentences as suggested.
  • Modified section on anecdotal health benefits. However, we would like to leave this section in the manuscript. Hence, this is what the participants tell us they think is happening and as such relevant (for example for future research).

Unfortunately we disagree with the reviewer on the use of inappropriate placebo. If anything, this is more ecologically valid comparison. 

Reviewer 2 Report

Thank you for the clarifications.

-L159: 12.7 g fibre per 100g dietary fibre. Is this a typo? Is it per 100g of product?

-L288-289. Without quantitative data with accompanying statistical analysis, this statement still is unsupported. This study does not appear to be designed to address differences based on weight status, so I recommend removing this.

Author Response

In response to the reviewer we have made the change to line 159. Indeed this is the product. In addition, we have removed the information in section 3.5.2.